# Impact of a 12-Week Hypocaloric Weight Loss Diet with Mixed Tree Nuts vs. Pretzels on Trimethylamine-N-Oxide (TMAO) Levels in Overweight Adults

**DOI:** 10.3390/nu17132137

**Published:** 2025-06-27

**Authors:** Onkei Lei, Jieping Yang, Hannah H. Kang, Zhaoping Li

**Affiliations:** 1Center for Human Nutrition, David Geffen School of Medicine, Department of Medicine, University of California, Los Angeles, Los Angeles, CA 90095, USAhkang03@g.ucla.edu (H.H.K.); 2Faculty of Education, University of Macau, Macao, China; 3Department of Medicine, VA Greater Los Angeles HealthCare System, Los Angeles, CA 90073, USA

**Keywords:** TMAO, gut microbiome, mixed tree nuts, choline intake, diet quality, hypocaloric diet, weight loss, HEI score, dietary intervention, cardiovascular health

## Abstract

Trimethylamine N-oxide (TMAO), a gut microbiome metabolite linked to cardiovascular health, can be influenced by dietary factors like choline intake and diet quality. This study compared the effects of mixed tree nuts (MTNs) and pretzels, as part of a 12-week hypocaloric weight loss diet, on TMAO levels and identified dietary predictors. Methods: Plasma samples from 95 overweight individuals consuming either 1.5 oz. of mixed tree nuts (MTNs, *n* = 56) or isocaloric pretzels (n = 39) daily for 12 weeks were analyzed. Nutritional data were collected at baseline and week 12 through dietary recall using the Automated Self-Administered 24 h Dietary Assessment Tool (ASA24), and the overall diet quality was assessed via the Healthy Eating Index (HEI) score. TMAO levels were determined and analyzed using linear mixed-effect models, adjusting for covariates. Wilcoxon signed-rank tests compared baseline and week 12 TMAO and weight. Multiple linear regression identified baseline predictors of TMAO. Results: Baseline demographics, anthropometric measures, HEI scores, and dietary choline intake were similar between the MTN and pretzel groups. A significant positive association was observed between baseline dietary choline and plasma TMAO levels (*p* = 0.012). The 12-week hypocaloric diet led to significant weight reduction in both groups (*p* < 0.01), but the magnitude of weight loss did not differ significantly between the MTN (−3.47 lbs) and pretzel (−4.25 lbs) groups (*p* = 0.18). Plasma TMAO levels decreased significantly in both groups (*p* < 0.01), but the between-group difference in reduction was not significant. (MTNs: −0.34 vs. pretzels: −0.37; *p* = 0.43). HEI scores and dietary choline intake remained unchanged, with no significant time–intervention interaction. Participants with low baseline HEI scores (≤53.72) had a more pronounced reduction in TMAO levels in the MTN group compared to the pretzel group (MTN: −0.54 vs. pretzel: −0.23; *p* = 0.045) over 12 weeks, despite similar weight loss. This difference was not observed in participants with higher HEI scores. Conclusions: The 12-week hypocaloric diet reduced body weight and plasma TMAO levels similarly in both MTN and pretzel groups. Participants with lower dietary quality saw a greater reduction in TMAO levels in the MTN group, suggesting MTNs may better modulate TMAO levels, especially for those with poorer baseline diets.

## 1. Introduction

Epidemiologic and experimental studies indicate that consuming tree nuts, including almonds, walnuts, pistachios, hazelnuts, cashews, and pecans, can lower the risk of cardiovascular diseases without contributing to weight gain, despite their energy density [1,2]. Their health benefits are often attributed to their rich composition of monounsaturated and polyunsaturated fatty acids, dietary fiber, plant protein, vitamins and minerals, as well as bioactive phytochemicals such as polyphenols and phytosterols. Tree nuts can impact body fat by regulating appetite, substituting less healthy nutrients, increasing diet-induced thermogenesis, providing usable energy, delivering anti-obesity benefits through bioactive compounds, and improving gut microbiome health [1,3]. In our previous randomized, controlled, two-arm study, we found that incorporating mixed tree nuts (MTNs) into a hypocaloric diet for 12 weeks resulted in increased satiety and reduced heart rate, with similar weight loss, and decreased diastolic blood pressure (DBP) compared to a pretzel control in overweight/obese healthy participants. The higher protein and fat content of MTNs compared to pretzels likely enhances satiety during hypocaloric weight loss diets [4]. Furthermore, our MTN intervention showed changes in both host and microbial tryptophan metabolism, while microbial composition remained largely unchanged [1,5].

Trimethylamine N-oxide (TMAO) is a metabolite produced by the liver from trimethylamine (TMA), which is generated by gut bacteria from dietary precursors such as choline (CHOLN), L-carnitine, and phosphatidylcholine (PtdCho) found in foods like red meat, eggs, and fish [5]. These dietary components are metabolized by gut microbiota into TMA, which is then absorbed into the bloodstream and transported to the liver. In the liver, TMA is oxidized by flavin-containing monooxygenases (FMOs) to form TMAO. This process highlights the intricate relationship between diet, gut microbiota, and liver function in the regulation of TMAO levels [6]. Elevated TMAO levels have been linked to insulin resistance and several sequelae of metabolic syndrome in humans, including cardiovascular, renal, and neurodegenerative diseases [7]. For instance, high TMAO levels are associated with an increased risk of atherosclerosis, a condition characterized by the buildup of fatty deposits in the arteries, which can lead to heart attacks and strokes. TMAO is thought to contribute to these conditions by promoting inflammation, enhancing cholesterol deposition in arterial walls, and increasing platelet aggregation, which can lead to thrombosis [7]. However, growing research also reports beneficial roles of TMAO, including antitumor activity and enhanced blood–brain barrier (BBB) integrity. Some studies suggest that TMAO may have protective effects against certain types of cancer. For instance, in pancreatic cancer, TMAO has been shown to inhibit tumor growth and metastasis by driving immune activation and boosting responses to immune checkpoint blockade [8]. Furthermore, TMAO has been shown to strengthen the BBB, which is crucial for protecting the brain from harmful substances and maintaining central nervous system homeostasis [9]. These findings indicate that TMAO’s role in human health is complex and may vary depending on the context and levels present in the body.

Diet can influence TMAO levels by modifying the gut microbiota, which affects the production of trimethylamine (TMA) from nutrients like choline and carnitine. In addition, phenolic compounds in foods can also impact the liver’s conversion of TMA to TMAO, potentially lowering TMAO levels and associated health risks [10]. MTNs, rich in phytochemicals such as carotenoids, phenolic acids, phytosterols, and polyphenolic compounds [11], may also affect TMAO production and metabolism. However, to our knowledge, no clinical trial has directly examined the effect of incorporating MTNs into a hypocaloric weight loss diet on circulating TMAO levels. This study aims to use blood samples and fecal microbiome data from a previous study to explore whether incorporating mixed tree nuts (MTNs) into a hypocaloric diet influences TMAO levels and their relationship with the gut microbiome. This could offer insights into dietary strategies for managing TMAO levels and reducing cardiovascular risk.

## 2. Materials and Methods

### 2.1. Study Design

We conducted this 24-week, randomized, controlled, open-label study in accordance with the guidelines set forth by the Human Subjects Protection Committee at UCLA. The clinical protocol received approval from the Internal Review Board (IRB) of UCLA. Prior to the commencement of the study, all participants provided written informed consent. This study is registered on ClinicalTrials.gov with the identifier NCT03159689 [1,4]. A total of 95 participants were randomly assigned to one of two intervention groups: mixed tree nuts (MTN, n = 56) and pretzel control (pretzels, n = 39). The 24-week study consisted of two phases: 12 weeks of weight loss followed by 12 weeks of weight maintenance. To investigate whether weight loss would lead to changes in TMAO levels, plasma samples at baseline and at the end of the 12-week weight loss phase were used in the current study.

### 2.2. Participants and Outcome Measures

Participants were healthy, overweight, or obese adults. Demographic, anthropometric, and biochemical measures were collected, including plasma lipids (total cholesterol, triglycerides, HDL cholesterol, and LDL cholesterol), plasma inflammatory markers high-sensitivity C-reactive protein (hs-CRP), interleukin 10 (IL-10), tumor necrosis factor-alpha (TNF-α), and monocyte chemoattractant protein-1 (MCP-1), and microbial diversity indices (Chao1 and Shannon) as previously described [1,4].

### 2.3. Dietary Intervention

All participants followed an individualized hypocaloric diet designed to induce a 500 kcal/day energy deficit as previously reported [1]. The MTN and pretzel groups consumed an isocaloric portion of either mixed tree nuts or refined carbohydrate (pretzels) daily. Participants received dietary counseling from a registered dietitian (RD) at baseline and biweekly throughout the study. At each session, participants were provided with meal plans, portion guidelines, and a checklist to record daily macronutrient intake. To ensure adherence, participants returned checklists and underwent dietary intake reviews. Nutritional intake was assessed at baseline and week 12 using the Automated Self-Administered 24 h Dietary Assessment Tool (ASA24), and the overall dietary quality was evaluated using the Healthy Eating Index (HEI-2015). The total HEI score translates habitual dietary intake into a composite score ranging from 0 to 100, with higher scores indicating greater adherence to the 2015–2020 Dietary Guidelines for Americans. Notably, 24 h dietary recalls at week 0 and week 12 were missing from 16 participants (Pretzels_Week 0: n = 37; Pretzels_Week 12: n = 34; MTNs_Week 0: n = 53; MTN_Week 12: n = 50). We included 32 participants in the pretzel group and 47 participants who completed dietary recall assessments at both week 0 and week 12.

### 2.4. TMAO Measurement via LC-MASS

Briefly, 25 µL plasma serum samples were spiked with TMAO-d9 Internal Standard (1 ug/mL, 20 uL), precipitated with 500 μL methanol overnight at −20 °C and centrifuged at 14,000× *g* for 15 min at 4 °C. Supernatant was dried using a SpeedVac evaporator (ThermoFisher Scientific, Waltham, MA, USA) and then resuspended in 50% methanol (100 uL) for analysis. LC coupled to electrospray ionization triple quadrupole MS (LC-ESI-MS/MS) at positive mode was used. The LC-ESI-MS/MS was performed on a 4.6 × 100 mm Phenomenex XB-C C18 column. Mobile phases were prepared for two pumps. Pump A was acetonitrile, and pump B was 0.1% formic acid/H2O. The gradient elution started with 2% A to 15% A in 10 min. Tandem MS spectra were automatically performed with argon as the collision gas (TMAO: *m*/*z* 76.2/58.2; TMAO-d9: 85.3/66.2). Concentrations were calculated by comparing the sample peak area with the ratio of the peak area to the internal standard peak area for LC/MS data using a method adapted from a previous publication [12].

### 2.5. Statistical Analysis

All statistical analyses were conducted using RStudio (version 2024.04.2+764; Posit Software, PBC, Boston, MA, USA). Data distribution was assessed using the Shapiro–Wilk test. Linear mixed-effect models were used to assess the main effects of intervention, time, and their interaction on TMAO levels, adjusting for age, sex, and race via the “lme4” package [13]. Subgroup analyses were conducted using three-way interaction models (intervention × time × HEI subgroup) to evaluate the impact of dietary quality. Baseline TMAO levels were analyzed for associations with cardiovascular disease risk factors using Spearman correlation, excluding unpaired data. Within-group statistical analysis was performed using the Wilcoxon signed-rank test. Cases with missing data were excluded from the analysis. Given the small number of pre-specified primary endpoints, *p*-value correction for multiple comparisons was not performed for these analyses.

### 2.6. Microbiome and Mediation Analysis

We previously performed per-feature testing in multivariate association with linear mixed-effect models established using MaAsLin2 in the R package to examine the potential association between species with time using previously reported microbiome data [4]. The current analysis was performed after filtering at a minimum abundance level of 0.0001 and a minimum prevalence of 0.5. Only significant associations between species over time, with *q* < 0.25 after applying a false discovery rate (FDR) correction, are reported. Mediation analysis was conducted to assess the extent to which the association between TMAO and time was mediated by differences in gut microbes. The analysis was performed using the mediation package in R [14]. Time was considered the primary exposure, microbes were the mediators, and TMAO was the outcome. Each mediator was tested separately, with age, gender, and race included as covariates, and adjustments made for subjects as a random effect.

## 3. Results

### 3.1. Association of Plasma TMAO Levels and Cardiovascular Risk Factors

We conducted a Spearman association analysis between baseline plasma TMAO concentrations and various clinical parameters linked to cardiovascular diseases (n = 95). Dietary intake data collected from the ASA24 recalls were used to calculate diet quality scores according to the Healthy Eating Index (HEI-2015) (n = 90). For the analysis, we examined associations between TMAO and both the total HEI score and several of its components, including added sugars (ADDSUG), fatty acids (FATTYACID), seafood and plant proteins (SEAPLANTPROT), and total protein (TOTPROT).

Our findings revealed that plasma TMAO was negatively associated with BMI (*r* = −0.304, *p* = 0.003) but positively associated with TNF-α (*r* = 0.209, *p* = 0.042) and hs-CRP (*r* = 0.209, *p* = 0.042), as well as dietary intake of CHOLN (*r* = 0.209, *p* = 0.041) and total protein food (*r* = 0.238, *p* = 0.024) (Figure 1).

### 3.2. Effects of Weight Loss Induced by a Hypocaloric Diet and the Incorporation of MTNs on Plasma TMAO Levels

Participants who consumed 1.5 oz. of MTNs or pretzels (PS) daily showed a significant decrease in plasma TMAO levels. In the MTN group, TMAO levels declined from 337.31 ± 31.53 ng/mL at baseline to 220.95 ± 15.8 ng/mL at week 12 (*p* = 0.002). Similarly, in the PS group, baseline levels were 399.3 ± 123.77 ng/mL and decreased to 218.31 ± 33.29 ng/mL at week 12 (*p* = 0.044). However, the reduction in TMAO levels did not differ between the MTN and pretzel groups over time (change from baseline: MTN: −114.26 ± 33.70 ng/mL vs. pretzel: 181.00 ± 131.12 ng/mL; *p* = 0.43) (Figure 2A).

### 3.3. Subgroup Analysis Using Dietary Quality (HEI)

A subgroup analysis was conducted using a median split of the baseline HEI-2015 scores from all 95 participants, a standard statistical method used to create equally sized comparison groups. Only participants with available baseline HEI-2015 scores (MTN: n = 47; pretzel: n = 32) were included in this subgroup analysis. Participants with HEI scores ≤53.72 were classified as HEI_low, while those with HEI scores >53.72 were classified as HEI_high. In the MTN group, 29 participants were classified as HEI_low, and 18 participants were classified as HEI_high. In the PS group, 15 participants were classified as HEI_low, and 17 participants were classified as HEI_high.

In the HEI_low subgroup, TMAO levels decreased from 364.43 (46.85) ng/mL at baseline to 232.46 ± 21.07 ng/mL at week 12 (*p* = 0.014). In the pretzel group, TMAO levels changed from a baseline level of 248.73 ± 38.84 ng/mL to 204.59 ± 42.38 ng/mL (*p* = 0.332). A significantly greater reduction in TMAO levels over 12 weeks was observed in the MTN group (−131.97 ± 49.71 ng/mL) compared to the pretzel group (−44.14 ± 59.44 ng/mL), with a *p*-value of 0.045 (Figure 2B).

In HEI_high participants, TMAO levels decreased from a baseline level of 272.15 ± 36.61 ng/mL to 193.4 ± 19.39 ng/mL at week 12 (*p* = 0.05). In the pretzel group, TMAO levels changed from 551.31 ± 236.9 ng/mL at baseline to 227.15 ± 54.9 ng/mL (*p* = 0.033). The difference in TMAO reduction between the two groups over time was not significantly different (Figure 2C).

### 3.4. TMAO Precursor Intake and Energy Intake

In this study, 24-h dietary recalls were collected and analyzed for HEI score at baseline and week 12. Although this study was designed as a hypocaloric diet weight loss study, the total energy intake was not significantly reduced in both the MTN and PS groups (Figure 3A). Dietary TMAO precursors, CHOLN and PtdCho, were not affected by the hypocaloric dietary intervention and remained unchanged between baseline and week 12 in the MTN and PS groups (Figure 3B,C). These findings suggest that the observed reduction in TMAO is unlikely to be due to a decreased intake of dietary TMAO precursors.

### 3.5. The Contribution of Microbiome to TMAO Changes

We examined the multivariable association between bacterial species and time. Only the relative abundance of Lachnospiraceae.UCG.010_NA was reduced significantly at week 12 compared to baseline, with *p* = 0.005 (*q* = 0.22). We performed a mediation analysis to investigate whether Lachnospiraceae.UCG.010_NA mediates the time-associated changes in TMAO levels during a 12-week hypocaloric weight loss dietary intervention. The results indicated that the average causal mediation effect (ACME) was not significant (estimate = −6.52, 95% CI = −27.3 to 11.4, *p* = 0.44), suggesting that *Lachnospiraceae.UCG.010_NA* did not significantly mediate the changes in TMAO levels.

## 4. Discussion

This study investigated the impact of a 12-week hypocaloric dietary intervention, with either MTNs or pretzels as snacks, on circulating levels of TMAO, a gut-derived metabolite linked to cardiovascular disease. Our primary finding was that the hypocaloric diet successfully induced weight loss and significantly reduced circulating TMAO levels in overweight and obese participants, with no overall difference between the MTN and pretzel groups. Secondly, our key novel finding emerged from a subgroup analysis based on baseline diet quality. Among participants with initially poor dietary habits (low HEI scores), the incorporation of MTNs into the hypocaloric treatment resulted in a more profound reduction in TMAO levels compared to the pretzel control, a benefit not observed in those with higher baseline diet quality.

We found that the hypocaloric dietary intervention did not significantly alter the intake of known dietary sources of TMAO, specifically choline and phosphatidylcholine. L-carnitine is another important dietary precursor of TMA. Although the HEI-2015 does not directly assess L-carnitine intake, trends can be inferred from components reflecting animal product consumption, such as total protein food and seafood and plant proteins. Since L-carnitine is predominantly found in animal products, particularly red meat, an increase in the total protein food component could suggest higher intake of L-carnitine-rich foods. During the intervention, the total protein food score significantly increased in the PS group (*p* = 0.041) and showed a trend toward increase in the MTN group (*p* = 0.08), whereas the seafood and plant protein score, which emphasizes seafood, legumes, nuts, and seeds, remained unchanged (Table 1). Therefore, the observed decrease in circulating TMAO levels during the hypocaloric dietary intervention is unlikely to be attributed to a reduction in dietary intake of TMAO precursors.

The positive associations between plasma TMAO concentrations and inflammatory markers, such as CRP and TNF-α, align with previous research highlighting TMAO’s role in promoting inflammation. Elevated TMAO levels have been linked to increased expression of pro-inflammatory cytokines, contributing to systemic inflammation and heightened cardiovascular risk [15]. However, the negative association between TMAO and BMI observed in our study contrasts with earlier findings, which often report a positive correlation between TMAO levels and obesity-related measures [16]. One difference is that the participants of the current study are healthy, overweight individuals from diverse racial backgrounds, unlike previous publications that focused more on overweight and obese individuals with metabolic syndrome. Metabolically healthy obese individuals are relatively protected against cardiometabolic diseases compared to obese individuals with metabolic syndrome [16,17,18]. This protection may involve factors such as the gut microbiome and adipose tissue plasticity. Investigating whether lower levels of TMAO contribute to this protection will be an interesting avenue for further study.

In addition, our study did not observe changes in microbial alpha diversity indices during the hypocaloric dietary intervention. Furthermore, no association between TMAO levels and microbial diversity indices was found, despite TMA being a microbial metabolite. This suggests that the reduction in TMAO levels observed in our study is not directly related to changes in the overall diversity of the gut microbiome. Interestingly, we noted that the abundance of Lachnospiraceae.UCG.010_NA was reduced during the hypocaloric dietary intervention. However, mediation analysis indicated that the reduction in TMAO levels was not regulated by Lachnospiraceae.UCG.010_NA. This highlights the complexity of microbial interactions and suggests that other factors may be contributing to the observed changes in TMAO levels.

It is known that the conversion of TMA to TMAO in the liver is a crucial step in determining the levels of TMAO in the body. Previous studies in mice have shown that extreme caloric restriction significantly increases FMO3 activity, the key hepatic enzyme responsible for converting TMA to TMAO [19]. Our current hypocaloric dietary intervention is a moderate caloric restriction, and whether this affects FMO3 activity needs further exploration. In subjects with poor dietary habits, incorporating MTNs led to a significant reduction in TMAO levels. Plant polyphenols have been shown to reduce the occurrence of atherosclerosis by modulating both the intestinal microbiota and hepatic FMO3 activity [20]. This suggests the potential of incorporating MTNs into hypocaloric dietary interventions for managing cardiovascular health.

This study is a well-monitored dietary intervention supported by regular counseling from a registered dietitian. It shows that a hypocaloric diet not only induces weight loss but is also associated with a reduction in plasma TMAO levels. Furthermore, incorporating MTNs as part of the hypocaloric diet also led to a similar reduction in TMAO. However, several limitations should be acknowledged. The open-label design, where participants were aware of their group assignment, may have introduced performance bias. Nutritional intake was assessed using the ASA24 self-administered dietary recall, a method prone to recall bias, which may explain the discrepancy between the observed weight loss and the lack of significant change in the reported energy intake. Additionally, the relatively small sample size limits the interpretation of subgroup analyses. Finally, as the study population consisted of healthy, overweight, or obese adults, the findings may not be generalizable to individuals with other metabolic conditions. Future studies involving larger, more diverse populations, particularly those with poor dietary habits, are needed to confirm these findings.

## 5. Conclusions

In summary, this 12-week hypocaloric diet incorporating either mixed tree nuts or pretzels as snacks leads to significant weight loss and reductions in plasma TMAO levels in overweight and obese adults. Notably, individuals with lower baseline diet quality experienced a greater reduction in TMAO when consuming mixed tree nuts compared to pretzels. The reduction in TMAO was not attributable to changes in dietary precursor intake or major shifts in gut microbiome diversity, highlighting the complexity of TMAO metabolism in response to dietary interventions. These findings suggest that including mixed tree nuts as part of a calorie-restricted diet may offer additional cardiovascular benefits, particularly for individuals with poorer baseline dietary habits. Further research in larger and more diverse populations is warranted to confirm these results and to elucidate the underlying mechanisms linking nut consumption, TMAO metabolism, and cardiometabolic risk.

## Figures and Tables

**Figure 1 nutrients-17-02137-f001:**
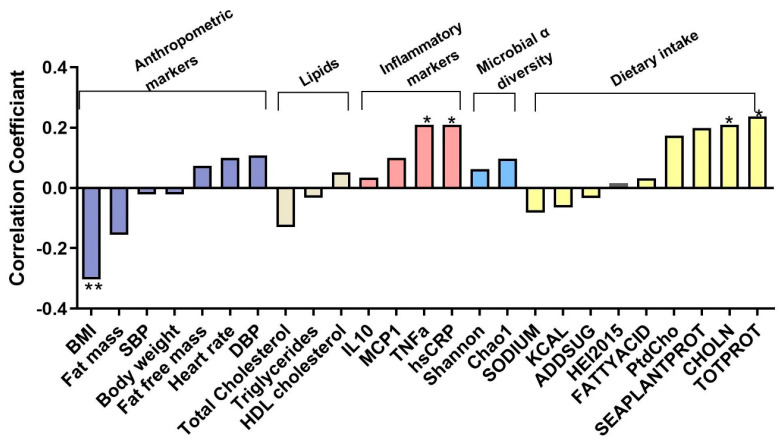
Spearman association analysis of baseline plasma TMAO concentrations with clinical and laboratory outcomes associated with cardiovascular diseases (n = 95). * Correlation is significant at *p* < 0.05. ** Correlation is significant at *p* < 0.01.

**Figure 2 nutrients-17-02137-f002:**
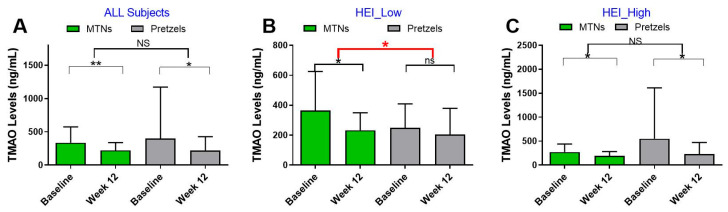
Plasma TMAO levels over 12 weeks: (**A**) all participants: MTN (n = 56), pretzel (n = 39); (**B**) HEI_low participants: MTN (n = 29), pretzel (n = 15); (**C**) HEI_high participants: MTN (n = 18), pretzel (n = 17). Data are presented as mean ± SD. The red line indicates a significant difference in the change from baseline between the MTN and pretzel groups. * *p* < 0.05, ** *p* < 0.01; NS, non-significant.

**Figure 3 nutrients-17-02137-f003:**
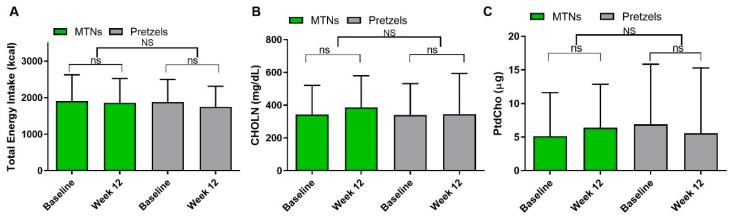
Dietary intake analysis and TMAO precursor levels following a 12-week hypocaloric diet intervention: daily (**A**) total energy intake, (**B**) CHOLN, and (**C**) PtdCho intake remained unchanged during intervention. NS, non-significant.

**Table 1 nutrients-17-02137-t001:** Changes in dietary intake and HEI-2015 components during the 12-week hypocaloric intervention. Components: total protein food and seafood and plant proteins (0–5; 0 = none, 5 = target intake), fatty acids, sodium, and added sugars (0–10; 0 = worst adherence, 10 = optimal adherence), and the total HEI-2015 score (0–100; higher = better diet quality). Values are mean (SD).

	Pretzel			MTN			*p* Interaction
	BL	W12	*p*	BL	W12	*p*	
*n*	37	34		53	50		
Energy (kcal)	1879.55 (618.52)	1750.71 (561.73)	0.190	1906.17 (720.38)	1858.02 (665.75)	0.929	0.451
CHOLN (mg)	339.18 (192.73)	344.86 (248.96)	0.846	343.39 (177.33)	387.26 (192.26)	0.267	0.518
PtdCho (mg)	6.89 (8.98)	5.58 (9.71)	0.497	5.14 (6.48)	6.4 (6.49)	0.321	0.28
Total Protein Food Component	3.86 (1.72)	4.6 (0.93)	0.041	4.46 (1.14)	4.8 (0.63)	0.080	0.239
Seafood and Plant Protein Component	0.64 (1.65)	1.17 (2.14)	0.188	0.9 (1.9)	0.9 (1.94)	0.708	0.199
Fatty Acid Component	5.95 (3.58)	7.04 (3.42)	0.284	5.58 (3.41)	8.22 (2.94)	0.000	0.127
Sodium Component	4.46 (3.65)	2.55 (2.82)	0.059	4.72 (3.62)	5.04 (3.38)	0.604	0.034
Added Sugar Component	8.16 (2.57)	9.18 (1.78)	0.056	8.31 (2.57)	8.93 (2.13)	0.100	0.546
HEI-2015 Total Score	51.41 (13.83)	53.27 (11.66)	0.400	49.64 (12.75)	57.91 (13.97)	0.005	0.095

## Data Availability

Data are contained within the article.

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
