# Peer review of "Impact of a 12-Week Hypocaloric Weight Loss Diet with Mixed Tree Nuts vs. Pretzels on Trimethylamine-N-Oxide (TMAO) Levels in Overweight Adults"

_nutrients, 2025, doi:10.3390/nu17132137_

Round 1
Reviewer 1 Report
Comments and Suggestions for Authors
This is an interesting research article with adequate novelty. Some points should be addressed.
- In Abstract, ASA24 and HEI scores should be explained.
- In Abstract (line 22), the word significant should be omitted since p-value is 0.18.
- Again, in Abstract (line 24), the word significantly should be omitted since p-value is 0.43.
- In the 1st paragraph of Introduction section, please provide a bit more details about tree nuts.
- Please also provide an explanaitionh mixed tree nuts reduced sateity.
- from a mechanistic point of you, please provide a bit description how TMAO exert antitumor effects and in what exactly types of cancer may exactly provide protective effects.
- A the last paragraph of Introduction and before the aim of the study, the authors should emhasize the literature gap that the aim to cover with the present study.
- In section 2.2, the authors should report whether anthrpometric data are measured data or self-reported data. If they are self-reported data which may lead to recal bias, then this issue should be reorted as a limitation of the study.
- In section 2.4, a relevant reference should be added.
- In section 2.5, did the authors used a normality test?
- The first tow paragraphs is a repetition of the results of the study. The authors should try to condense these two paragraphs into one paragraph by reporting only the mosy significants results.
- At the end of the Discussion section, the authors should add a paragraph with the Strengths and the Limitations of the study.
- A Conclusion section is missing.
Author Response
Comments 1: In Abstract, ASA24 and HEI scores should be explained.
Response 1: Thank you for your comment. We have revised the Abstract (lines 16-18) to define both acronyms on first use:
Revised text in Abstract:
Lines 16-18:
“Nutritional data were collected at baseline and week 12 with the Automated Self-Administered 24-hour (ASA24) dietary recall, and overall diet quality was assessed via the Healthy Eating Index (HEI) score”
Comments 2: In Abstract (line 22), the word significant should be omitted since p-value is 0.18.
Response 2: Thank you for your comment. We agree the original phrasing was unclear. Our intent was to convey that both groups experienced significant weight loss from baseline, but the difference between groups was not statistically significant (p = 0.18). We have revised the Abstract to clarify this:
Revised text in Abstract:
Line 24-25:
"The 12-week hypocaloric diet led to significant weight reduction in both groups (p < .01), but the magnitude of weight loss did not differ significantly between the MTN (-3.47 lbs) and Pretzel (-4.25 lbs) groups (p = 0.18)."
Comments 3: Again, in Abstract (line 24), the word significantly should be omitted since p-value is 0.43.
Response 3: Thank you for your comment. We agree the sentence was unclear. Our aim was to convey two findings: (1) plasma TMAO levels significantly decreased within each group (MTN: p = 0.002; Pretzel: p = 0.044), and (2) the between-group difference was not significant (p = 0.43). We have revised the Abstract to clarify this:
Revised text in Abstract:
Line 26-27:
"Plasma TMAO levels decreased significantly within both the MTN and Pretzel groups (p < .01), but the between-group difference in reduction was not significant. (MTNs: -0.34 vs. Pretzels: -0.37; p = 0.43).”
Comments 4: In the 1st paragraph of Introduction section, please provide a bit more details about tree nuts.
Response 4: We have added health attributes of tree nuts in the 1st paragraph.
Revised text in Introduction:
Line 40-59:
“Epidemiologic and experimental studies indicate that consuming tree nuts, including almonds, walnuts, pistachios, hazelnuts, cashews, and pecans, can lower the risk of cardiovascular diseases without contributing to weight gain, despite their energy density (1,2). Their health benefits are often attributed to their rich of composition of monosatu-rated and polyunsaturated fatty acids, dietary fiber, plant protein, vitamins and minerals, as well as bioactive phytochemicals such as polyphenols and phytosterols.”
Comments 5: Please also provide an explanaitionh mixed tree nuts reduced sateity.
from a mechanistic point of you, please provide a bit description how TMAO exert antitumor effects and in what exactly types of cancer may exactly provide protective effects.
Response 5: We added the following sentence to (1) suggest a possible mechanism for increased satiety and (2) added the exact cancer type:
Added text in Introduction:
Line 66-68:
“The higher protein and fat content of MTNs compared to pretzels likely enhances satiety during hypocaloric weight loss diets (4).”
Line 88-90:
“Some studies suggest that TMAO may have protective effects against certain types of cancer. For instance, in pancreatic cancer, TMAO has been shown to inhibit tumor growth and metastasis by driving immune activation and boosting responses to immune checkpoint blockade (7)”
Comments 6: A the last paragraph of Introduction and before the aim of the study, the authors should emhasize the literature gap that the aim to cover with the present study.
Response 6: Thank you for your comment. We have added a sentence to state the literature gap:
Added text in Introduction:
Line 99-101:
“However, to our knowledge, no clinical trial has directly examined the effect of incorporating MTNs into a hypocaloric weight-loss diet on circulating TMAO levels.”
Comments 7: In section 2.2, the authors should report whether anthrpometric data are measured data or self-reported data. If they are self-reported data which may lead to recal bias, then this issue should be reorted as a limitation of the study.
Response 7: Thank you for your comments, all anthropometric measures were directly measured by trained research staff during study visits and were not self-reported. Regarding nutritional intake, data were collected using the Automated Self-Administered 24-hour (ASA24) Dietary Assessment Tool. While the data was collected with registered dietitian’s guidance, by its nature, this method relies on participant self-reporting and is subject to potential recall and reporting bias, which is a well-established limitation in nutritional science. To make this limitation more explicit as per the reviewer's suggestion, we will integrate the relevant limitation into a new paragraph about strengths and limitations at the end of the Discussion section as suggested in the Comment 11.
Revised text in Discussion:
Line 410-413:
“Nutritional intake was assessed using the ASA24 self-administered dietary recall, a method prone to recall bias, which may explain the discrepancy between the observed weight loss and the lack of significant change in reported energy intake.”
Comments 8: In section 2.4, a relevant reference should be added.
Response : Thank you. We have added the reference in Section 2.4 (line 161).
Comments 9: In section 2.5, did the authors used a normality test?
Response 9: Thank you for your comment. We ran Shapiro-Wilk tests on our key outcomes. While key variables such as weight were normally distributed (p > .15), the main outcome, TMAO, and several secondary biomarkers were found to be non-normal (p < .01). Consequently, our analytical strategy relied on mixed-model residual diagnostics for inferential testing, and employing non-parametric tests (e.g. Wilcoxon, Spearman) for analyses where normality assumptions were not met.
To ensure full transparency regarding this process, we will add a statement of our normality testing.
Added text in Section 2.5:
Line 164:
“Data distribution was assessed using the Shapiro-Wilk test.”
Comments 10: The first tow paragraphs is a repetition of the results of the study. The authors should try to condense these two paragraphs into one paragraph by reporting only the mosy significants results.
Response 10: Thank you for this suggestion. We’ve condensed the opening of the Discussion to highlight the most critical findings.
Original text:
“A hypocaloric dietary intervention led to weight loss, often associated with a re-duction in cardiovascular risk factors. TMAO is a novel cardiovascular disease risk factor (5). We have previously reported that a moderate hypocaloric dietary intervention (-500 calories daily) led to significant weight loss, and using energy-dense and polyphenol-rich MTNs as part of a hypocaloric weight loss diet resulted in similar weight loss. This hypocaloric dietary intervention showed signs of reduced cardiovascular risk by lowering blood pressure and heart rate, specifically in the MTNs group (1). Therefore, we test the hypothesis that a hypocaloric diet with MTNs impacts TMAO levels, which could provide insights into its potential benefits for cardiovascular health.
Our study yielded several key findings. First, we observed a significant reduction in circulating TMAO levels among participants following a hypocaloric diet-induced weight loss, with the incorporation of MTN into the hypocaloric diet having no impact on this reduction. Second, our exploratory analysis showed that participants with initially poor dietary habits experienced a more profound reduction in TMAO levels when MTNs were incorporated into a hypocaloric treatment compared to the pretzel control. In contrast, participants with good dietary habits showed similar reductions in TMAO levels in both groups over hypocaloric dietary intervention.”
Revised text:
Line 290-299:
“This study investigated the impact of a 12-week hypocaloric dietary intervention, with either MTNs or pretzels as snack, on circulating levels of TMAO, a gut-derived metabolite linked to cardiovascular disease. Our primary finding was that the hypocaloric diet successfully induced weight loss and significant reduced circulating TMAO levels in overweight and obese participants, with no overall difference between the MTN and pretzel groups. Secondly, our key novel finding emerged from a subgroup analysis based on baseline diet quality. Among participants with initially poor dietary habits (low HEI scores), the incorporation of MTNs into the hypocaloric treatment resulted in a more profound reduction in TMAO levels compared to the pretzel control, a benefit not observed in those with higher baseline diet quality.”
Comments 11: At the end of the Discussion section, the authors should add a paragraph with the Strengths and the Limitations of the study.
Response 11: Thank you for this suggestion. The new paragraph detailing the study's strengths and limitations has been added to the end of the Discussion section as suggested.
Line 405-418:
“This study is a well-monitored dietary intervention supported by regular counseling from a registered dietitian. It shows that a hypocaloric diet not only induces weight loss but is also associated with a reduction in plasma TMAO levels. Furthermore, incorporating MTNs as part of the hypocaloric diet also lead to similar reduction of TMAO. However, several limitations should be acknowledged. The open-label design, where participants were aware of their group assignment, may have introduced performance bias. Nutritional intake was assessed using the ASA24 self-administered dietary recall, a method prone to recall bias, which may explain the discrepancy between the observed weight loss and the lack of significant change in reported energy intake. Additionally, the relatively small sample size limits the interpretation of subgroup analyses. Finally, as the study population consisted of healthy, overweight or obese adults, the findings may not be generalizable to individuals with other metabolic conditions. Future studies involving larger, more diverse populations, particularly those with poor dietary habit would be needed to confirm these findings.”
Comments 12: A Conclusion section is missing.
Response 12: Please see out response to comment 11.
Reviewer 2 Report
Comments and Suggestions for Authors
This manuscript describes an analysis of data previously collected from a study conducted several years ago. This is a new analysis investigating the impact of mixed tree nuts on TMAO. The introduction provides the necessary background to understand this additional analysis. I have a few comments.
- Is the data normally distributed? There is no mention in the statistical section.
- Line 18: Paired t-tests are mentioned. Was p corrected for multiple comparisons?
- Line 152-155 state, “Dietary quality is assessed using the Healthy Eating Index (HEI) total score and its components: added sugars (ADDSUG), fatty acids (FATTYACID), seafood and plant proteins (SEAPLANTPROT), and total protein (TOTPROT).” Added sugars, fatty acids, seafood and plant proteins, and total protein are components of the Healthy Eating Index, which is a measure of dietary quality. I am not sure what is being stated here. As stated, it sounds like a different diet quality measure is being calculated. Table 1 presents the HEI score, along with these components.
- Section 3.3: Why was 53.72 selected as the cutoff? No rationale is provided. Why not select the average HEI score for the American diet?
- Section 3.3: The numbers don’t add up. There are 56 participants in the MTN group, but MTN HEI_low = 31 and MTN HEI_high = 21, which adds up to 51. There are 39 in the PS group, but PS-low = 17 and PS_high = 20, which adds up to 37. What happened to six participants?
- Figure 2: The legend labels should be larger. They are hard to read.
- The abbreviations CHOLN and PtdCho are never defined.
- Table 1: There should be more space between the second line of energy and the first line of choline. They run together.
- Table 1: For HEI score, the range would be helpful too.
- Line 280: TMAo should be TMAO. Should levels have a period instead of a comma?
Author Response
Comments 1: Is the data normally distributed? There is no mention in the statistical section.
Response 1: Thank you for your comment. We ran Shapiro-Wilk tests on our key outcomes. While key variables such as weight were normally distributed (p > .15), the main outcome, TMAO, and several secondary biomarkers were found to be non-normal (p < .01). Consequently, our analytical strategy relied on mixed-model residual diagnostics for inferential testing, and employing non-parametric tests (e.g., Wilcoxon, Spearman) for analyses where normality assumptions were not met.
To ensure full transparency regarding this process, we will add a statement of our normality testing.
Added text in Section 2.5:
Line 164:
“Data distribution was assessed using the Shapiro-Wilk test.”
Comments 2: Line 18: Paired t-tests are mentioned. Was p corrected for multiple comparisons?
Response 2: Thank you for your comment. We corrected the typo “paried t-test” and clarified that the Wilcoxon signed-rank test was used, as stated in the Methods section (line 163). A formal correction for multiple comparisons was not applied to these specific tests. Our rationale was that for a small number of pre-specified primary outcomes (TMAO and weight), corrections can be overly conservative and increase the risk of false negatives (Type II errors). To improve clarity, we have refined the manuscript to better reflect the tests used and the rationale.
Added text:
Line 176-177:
“Given the small number of pre-specified primary endpoints, p-value correction for multiple comparisons was not performed for these analyses.”
Comments 3: Line 152-155 state, “Dietary quality is assessed using the Healthy Eating Index (HEI) total score and its components: added sugars (ADDSUG), fatty acids (FATTYACID), seafood and plant proteins (SEAPLANTPROT), and total protein (TOTPROT).” Added sugars, fatty acids, seafood and plant proteins, and total protein are components of the Healthy Eating Index, which is a measure of dietary quality. I am not sure what is being stated here. As stated, it sounds like a different diet quality measure is being calculated. Table 1 presents the HEI score, along with these components.
Response 3: Thank you for your comment. To clarify, raw dietary intake data were first collected from participants using the ASA24 24-hour dietary recall tool. This raw data was then processed to calculate diet quality scores based on the Healthy Eating Index (HEI-2015) framework. For our analysis, we examined associations between TMAO and both the overall HEI total score and several of its components. To accurately reflect this process, we have rewritten the sentence in Section 3.1.
Revised text in Section 3.1:
Line 194-199:
“Dietary intake data collected from the ASA24 recalls were used to calculate diet quality scores according to the Healthy Eating Index (HEI-2015) (n = 90). For the analysis, we examined associations between TMAO and both the HEI Total Score and several of its components, including added sugars (ADDSUG), fatty acids (FATTYACID), seafood and plant proteins (SEAPLANTPROT), and total protein (TOTPROT).”
Comments 4: Section 3.3: Why was 53.72 selected as the cutoff? No rationale is provided. Why not select the average HEI score for the American diet?
Response 4: Thank you for this excellent question. We chose not to use the national average HEI score for subgroup analysis because HEI scores vary widely across different racial, cultural, and demographic groups. Given this diversity and the specific characteristics of our study population, applying a single national average would not accurately reflect dietary quality within our cohort. Instead, we used a median split of HEI scores within our sample to better capture relative differences in diet quality and ensure more meaningful subgroup comparisons.
Revised text:
Line 214-216:
“A subgroup analysis was conducted using a median split of the baseline HEI-2015 scores from all 95 participants, a standard statistical method used to create equally sized comparison groups”
Comments 5: Section 3.3: The numbers don’t add up. There are 56 participants in the MTN group, but MTN HEI_low = 31 and MTN HEI_high = 21, which adds up to 51. There are 39 in the PS group, but PS-low = 17 and PS_high = 20, which adds up to 37. What happened to six participants?
Response 5: Thank you for identifying this numerical inconsistency. We have revised Section 3.3 to clarify that the analysis was performed only on participants with complete dataset at baseline and week 12 and to report the corrected numbers.
Revised text:
Line 194-197:
“This subgroup analysis included only participants with complete HEI-2015 scores at both baseline and week 12 (MTN: n = 47; Pretzels: n = 32). Participants with HEI scores ≤ 53.72 were classified as HEI_low, while those with HEI scores > 53.72 were classified as HEI_high. In the MTN group, 29 participants were classified as HEI_low and 18 participants were classified as HEI_high. In the PS group, 15 participants were classified as HEI_low and 17 participants were classified as HEI_high.”
Comments 6: Figure 2: The legend labels should be larger. They are hard to read.
The abbreviations CHOLN and PtdCho are never defined.
Response 6: We have revised Figure 2 to increase the font size of the legend labels for better readability. Also, we have now defined Choline (CHOLN) and Phosphatidylcholine (PtdCho) upon their first appearance in the Introduction. To ensure clarity and consistency throughout the manuscript, we have also updated the labels in Table 1.
Revised text:
Line 73:
“Trimethylamine N-oxide (TMAO) is a metabolite produced by the liver from tri-methylamine (TMA), which is generated by gut bacteria from dietary precursors such as choline (CHOLN), L-carnitine, and phosphatidylcholine (PtdCho) found in foods like red meat, eggs, and fish (5).”
Comments 7: Table 1: There should be more space between the second line of energy and the first line of choline. They run together.
Response 7: Thank you for pointing out this formatting issue. We have revised the formatting of Table 1 to improve readability as requested.
Comments 8: Table 1: For HEI score, the range would be helpful too.
Response 8: Thank you for this helpful suggestion. We have revised the caption in Table 1 (Line 256).
Revised text:
Line 274-276:
“Table 1. Changes in dietary intake and HEI-2015 components during the 12-week hypocaloric intervention. Components: Total Protein Foods & Seafood and Plant Proteins (0-5; 0 = none, 5 = target intake), Fatty Acids & Sodium & Added Sugars (0-10; 0 = worst adherence, 10 = optimal adherence), and HEI-2015 Total Score (0-100; higher = better diet quality). Values are mean (SD).”
Comments 9: Line 280: TMAo should be TMAO. Should levels have a period instead of a comma?
Response 9: revised.
Round 2
Reviewer 1 Report
Comments and Suggestions for Authors
The authors have significantly improved their manuscript.
Reviewer 2 Report
Comments and Suggestions for Authors
The authors have responded to my comments. I feel the manuscript is ready for publication.